# Theoretical and Experimental Aspects of Sodium Diclofenac Salt Release from Chitosan-Based Hydrogels and Possible Applications

**DOI:** 10.3390/gels9050422

**Published:** 2023-05-17

**Authors:** Loredana Maria Himiniuc, Razvan Socolov, Irina Nica, Maricel Agop, Constantin Volovat, Lacramioara Ochiuz, Decebal Vasincu, Ana Maria Rotundu, Iulian Alin Rosu, Vlad Ghizdovat, Simona Ruxandra Volovat

**Affiliations:** 1Department of Obstetrics and Gynecology, “Grigore T. Popa” University of Medicine and Pharmacy, 700115 Iasi, Romania; 2Department of Odontology-Periodontology, Fixed Prosthesis, “Grigore T. Popa” University of Medicine and Pharmacy, 700115 Iasi, Romania; 3Department of Physics, “Gheorghe Asachi” Technical University of Iasi, 700050 Iasi, Romania; 4Romanian Scientists Academy, 050094 Bucharest, Romania; 5Department of Medical Oncology-Radiotherapy, “Grigore T. Popa” University of Medicine and Pharmacy, 16 University Street, 700115 Iasi, Romania; 6Faculty of Pharmacy, “Grigore T. Popa” University of Medicine and Pharmacy, 700115 Iasi, Romania; 7Department of Biophysics, Faculty of Dental Medicine, “Grigore T. Popa” University of Medicine and Pharmacy, 700115 Iasi, Romania; 8Faculty of Physics, “Alexandru Ioan Cuza” University of Iasi, 700506 Iasi, Romania; 9Department of Biophysics and Medical Physics, “Grigore T. Popa” University of Medicine and Pharmacy, Iasi 700115, Romania

**Keywords:** chitosan, sodium diclofenac salt, drug-delivery systems, nanomedicines, endometrial damage, intrauterine adhesions, periodontal disease, cancer, multifractality

## Abstract

Two formulations based on diclofenac sodium salt encapsulated into a chitosan hydrogel were designed and prepared, and their drug release was investigated by combining in vitro results with mathematical modeling. To understand how the pattern of drug encapsulation impacted its release, the formulations were supramolecularly and morphologically characterized by scanning electron microscopy and polarized light microscopy, respectively. The mechanism of diclofenac release was assessed by using a mathematical model based on the multifractal theory of motion. Various drug-delivery mechanisms, such as Fickian- and non-Fickian-type diffusion, were shown to be fundamental mechanisms. More precisely, in a case of multifractal one-dimensional drug diffusion in a controlled-release polymer–drug system (i.e., in the form of a plane with a certain thickness), a solution that allowed the model’s validation through the obtained experimental data was established. The present research reveals possible new perspectives, for example in the prevention of intrauterine adhesions occurring through endometrial inflammation and other pathologies with an inflammatory mechanism background, such as periodontal diseases, and also therapeutic potential beyond the anti-inflammatory action of diclofenac as an anticancer agent, with a role in cell cycle regulation and apoptosis, using this type of drug-delivery system.

## 1. Introduction

Non-steroidal anti-inflammatory drugs (NSAIDs), including diclofenac sodium salt (DCF), represent the first therapeutic choice in the management of acute and chronic pain and inflammatory diseases [1,2]. Moreover, it was reported that DCF has antitumor properties in various cancers, such as glioma, fibrosarcoma, colorectal, and pancreatic cancer [3,4,5,6]. This is due to its ability to inhibit both cyclooxygenases—COX-1 and COX-2—two enzymes that synthesize prostaglandins (PGs). When diclofenac sodium binds to COX isoenzymes, it leads to the inhibition of prostanoid (PGE_2_, PGD_2_, thromboxane A_2_, and prostacyclin I_2_) synthesis. Prostaglandin E2 (PGE_2_) is the prostanoid most implicated in inflammatory processes, and its inhibition by NSAIDs represents the key mechanism behind their anti-inflammatory and analgesic properties. In the gynecological field, DCF is frequently used to treat endometriosis, ovarian cancer, pelvic inflammatory disease, endometrial inflammation, Asherman syndrome, abnormal uterine bleeding, or perioperative pain [7]. Both COX enzymes are expressed in the uterine epithelium in different periods in the first trimester of pregnancy [8]. Moreover, it has been shown that these two isoenzymes have important roles in the mechanism of endometrial pathology. The immune expression of COX-2 was found to be high in acute endometritis compared with chronic endometritis or normal endometrium [9].

Consequently, DCF is used in different treatment applications in several formulations, such as parenteral, oral, and topical administration. However, its oral administration has various dose-dependent gastrointestinal, cardiovascular, and renal side effects. To overcome this drawback, novel technologies are focused on the improvement of pharmacological properties, new delivery pathways through nanotechnology, and co-administration with agents that offer gastric protection in order to improve the drug’s tolerability and enhance its clinical indications [10]. One possible path towards finding an effective treatment using DCF is the employment of drug-delivery systems. More specifically, drug delivery is a modern research direction of precise medicine, targeting the enhancement of drugs’ therapeutic effects in humans or animals [11,12]. Many strategies have been investigated to this end, resulting in a diverse range of new methods and materials. Among them, the hydrogel-based formulations are at the forefront, due to their high similarity with normal tissues and their ability to be easily molded, filling any space. Moreover, their porous morphology can ensure gas exchange and drug diffusion [13]. Among the polymers used for obtaining hydrogels, those of natural origin are preferred due to their biocompatibility and biodegradability. This is the case of chitosan, which besides intrinsic biocompatibility and biodegradability has a plethora of valuable bioactive properties, making it a promising candidate for the design of multifunctional materials with synergic activity [14,15,16,17]. 

Amongst the routes investigated for chitosan hydrogelation, that of crosslinking with natural aldehydes has revealed potential for real-life applications [15,16,17]. It was demonstrated that hydrogels prepared from chitosan and monoaldehydes can easily accommodate drugs, leading to formulations with prolonged release, which can be controlled by the crosslinking density [18,19,20,21,22,23,24,25,26]. This suggests the crosslinking of chitosan with monoaldehydes in the presence of a drug as a promising route for preparing formulations for local drug delivery.

In the light of these literature data, the present research investigated the drug-release behavior of two formulations prepared by the in situ crosslinking of chitosan with 2-hydroxybenzaldehyde in the presence of DCF. The drug delivery was monitored in vitro, and a mathematical model was created to understand the release kinetics for the further development of formulations for clinical applications.

## 2. Results and Discussion

Two formulations were prepared by the in situ encapsulation of DCF during the hydrogelation of chitosan with 2-hydroxybenzaldehyde, in two different molar ratios of functionalities (i.e., amine and aldehyde groups: D1.5 and D2 formulations). The total reaction of aldehyde with chitosan was demonstrated by the FTIR spectra of the corresponding xerogels, which showed the presence of the characteristic imine band at 1628 cm^−1^ and the absence of the characteristic band of aldehyde at 1670 cm^−1^ (Figure 1) [20]. Moreover, the presence of the drug was demonstrated by the presence of its specific absorption band, the vibration of the carboxyl group around 1575 cm^−1^, which appeared as a shoulder overlapped with the vibration band of amino units of chitosan [18]. The hydrogel state of the formulations was firstly assessed by inverted tube test, which showed the hydrogel formation after a few minutes from the components’ mixture (Figure 2b,c). It was further demonstrated by the rheological measurements, which showed the elastic modulus (G’) to be higher than viscous modulus (G’’) (Figure 2a) [21]. The elastic modulus of D1.5 was higher than that of D2, indicating a more rigid network, in agreement with a higher crosslinking density. The hydrogels swelled very fast, reaching mass equilibrium in less than 1 h, with an average swelling degree of 19 g/g for D1.5 and 27 g/g for D2 (Figure 2d,e).

To understand the drug-release mechanism, it is important to establish the drug’s pattern of encapsulation into the formulations. To this end, the formulations were investigated by SEM and POM.

POM images showed strong birefringence reassembling a continuous banded texture specific for a layered supramolecular architecture, which drives the hydrogelation of chitosan with monoaldehydes (Figure 3a,b) [15,16,17,20]. No DCF crystals were detected, indicating that the drug was anchored into the hydrogel matrix at a sub-micrometric level, under the detection limit of the microscope [27]. D1.5 showed a finer texture, with richer details, in agreement with a higher number of layered clusters, and consequently a higher crosslinking density [20,28]. Considering the chemical structure of DCF and chitosan, it can be envisaged that DCF was anchored in the chitosan-based matrix by H-bonds between the H atoms of the OH or NH_2_ units of chitosan and electronegative chlorine and oxygen atoms of DCF [18]. 

The morphology of formulations impacts the efficiency of drug delivery, as it is known that a porous structure favors a drug’s mobility through it. The SEM images displayed a porous morphology for both formulations, with interconnected pores and thick walls (Figure 3). The higher crosslinking density of D1.5 was reflected in smaller pores, while the lower crosslinking density of D2 led to larger pores [20,23]. Once again, no DCF crystals were detected in the SEM images; no geometrical shapes characteristic of DCF crystals were distinguished, neither in the pores nor in the pore walls. This confirmed that the DCF was embedded into the pore walls at the sub-micrometric level, most probably by H-bonding to chitosan.

The in vitro DCF release from the D1.5 and D2 formulations was investigated over 10 days, showing prolonged release, with slight differences between the two samples (Figure 4). The amount of DCF released was around 40% in the first 24 h and reached around 70% after 10 days. Overall, the rate of DCF release from the D1.5 formulation was slightly lower than that for D2, in agreement with its higher crosslinking density, which caused slower swelling and, consequently, slower diffusion of the DCF molecules. Nevertheless, the curves showed a constant rate of DCF release, with no significant burst effect compared to other chitosan-based formulations [14]. For instance, recent studies reported that chitosan–alginate hydrogels promoted much faster DCF release, realizing 60% drug release in 6 h. [28]. Prolonged DCF release was achieved when synthetic components such as acrylamide, acrylic acid, and acrylonitrile monomers were applied to prepare chitosan-based hydrogels, reaching around 40% after 6 h, still higher than that reported in this paper [29]. On the other hand, hydrogels prepared from chitosan binary grafted with poly(N-vinylcaprolactam) and poly(acrylic acid) showed the almost total release of DCF in only 3 h (Figure 5a,b) [30]. A ternary hydrogel based on β-glycerol phosphate, genipin, and chitosan, after a composition optimization, achieved complete DCF release after 8 h (Figure 5a,b). A somewhat similar kinetic release of DCF has been reported for systems prepared from mesoporous silica/PEI microspheres coated with carboxymethyl cellulose/chitosan [31]. This comparison highlights the advantage of using a monoaldehyde as a crosslinker, to delay the DCF release from chitosan-based hydrogels. It can be assumed that this crosslinking led to a tight network, which controlled the diffusion of the drug and, ultimately, its release. Moreover, the physical forces that can be developed between the drug and chitosan network can play a major role in delaying the DCF release [18,31].

### Correspondence of the Theoretical Model with the Experimental Data

Following the methodology from [26], for an identical structural system, the solution of the multifractal diffusion equation can be obtained in the form
(1)f=ρtρ∞=2σtd212π−12+∑n=1∞−1nerfcnd2σt12

For small time scales, reachable through experiments, the second term of Equation (1) disappears, and it leads to the following relation:(2)$$\frac{\rho_{t}}{\rho_{\infty }}=2{\left(\frac{\sigma t}{d^{2}}\right)}^{\frac{1}{2}}=const\cdot t^{\frac{1}{2}}\Leftrightarrow \frac{M_{t}}{M_{\infty }}=2{\left(\frac{\sigma t}{d^{2}}\right)}^{\frac{1}{2}}=const\cdot t^{\frac{1}{2}}$$
which is actually the Higuchi equation, $\frac{M\left(t\right)}{M\left(\infty \right)}=k_{H}t^{\frac{1}{2}}$, one of the best-known equations used in modeling release kinetics, with $M_{t}$ being the amount of drug released in the time interval t and $M_{\infty }$ the amount released in an infinite time interval, which will correspond, in fact, to the drug initially loaded into the polymer matrix.

In Figure 6, the drug-release mass is represented, depicted in percentages (%) as a function of time (we took into consideration only the first twenty terms of Equation (1)). A fast increase in the drug-release mass was seen in the first 24 h, followed by a quasi-saturation region, reaching a maximum of approximately 70% of the total drug mass. The multifractal model was used to fit the empirical data. The simulations were performed by setting free all the parameters, so that, through the fitting procedure, the fractal degree of the system could be extracted. The fractal degree (fractalization degree) is a parameter characterizing the representation of the drug-release phenomena in a multifractal space. It can be observed that higher values of the fractal degrees lead to a higher concentration of drug mass released in the system. Here, the values for the D1.5 and D2 systems are quite close. This is reflected by the data from Figure 6, where we can observe that the SEM image of the formulations indicate a similar morphology. The fractal degree is related to the interchange between the formulation matrix and the release medium, and it depends on a wide series of parameters, such as morphology, hydrophilicity, drug-mass concentration, and recipe for complex compound formation. The model is seen here to fit well with the data.

Inflammation plays a key role in injured endometrium. Due to the variety of NSAID side effects that other administration methods cause, we found this model of drug-delivery system to be a promising therapeutic strategy that may be very attractive in inflammation-related endometrial pathology and intrauterine adhesions. Both NSAID COX enzymes are expressed in the uterus epithelium, and it is shown that these two isoenzymes play a key role in the mechanism of endometrial pathology. The immune expression of COX-2 was found to be the greatest in acute endometritis compared with chronic endometritis or normal endometrium [8,9]. Furthermore, other pathologies that present properties beyond an inflammatory mechanism may also be good candidates for this drug-delivery system strategy. It is believed that the lipopolysaccharides and DNA contained in bacterial dental plaque detected in patients with periodontitis are responsible for the production of proinflammatory cytokines and chronic systemic inflammation. In addition, these proinflammatory cytokines can cause the recruitment of hyperresponsive neutrophils, which increase the production of reactive oxygen species, but further studies are needed to establish a correlation between these pathologies [29]. There is a clinical demand for new, highly efficient treatments for both intrauterine adhesion prevention and periodontal disease, as well as in cancer treatment strategies. Selectively targeting drugs to the inflamed endometrium may improve the therapeutic results and lessen systemic toxicity.

Chitosan has excellent bio- and muco-adhesivity, which is beneficial for gynecological applications. Investigations on combinations of chitosan and intrauterine devices showed efficacy in reducing adhesions and improving clinical outcomes in patients with moderate-to-severe hysteroscopic adhesiolysis [32]. Poloxamer/chitosan hydrogels have proven suitability for drug delivery in the vagina [33]. Chitosan as a component of mucoadhesive drug delivery systems for the treatment of vaginal infectious diseases demonstrated efficacy due to its intrinsic antimicrobial activity, which did not interfere with the effectiveness of the drugs [34]. Chitosan also has proven advantages as matrix for co-delivery of anticancer drugs and an immunoadjuvant for successful chemo-immunotherapy [35]. Moreover, chitosan hydrogel formulations have been proven to be an important tool to improve DCF bioavailability for the treatment of endometriosis pain [36].

## 3. Conclusions

Two chitosan-based formulations, containing DCF as a model drug, were prepared by in situ crosslinking with o-hydroxyl benzaldehyde. POM and SEM microscopy indicated that the DCF drug was embedded into the hydrogel pore walls at the sub-micrometric level. The in vitro drug release of DCF showed prolonged delivery, reaching 40% in the first 24 h. A new theoretical model to describe drug-delivery processes was proposed, based on the multifractal paradigm of motion. From this perspective, using the multifractal hydrodynamic-type equations, the description of drug-delivery processes becomes reducible, through the synchronization of drug-delivery dynamics, both at differentiable and at non-differentiable scale resolutions, to the diffusion equation of a multifractal type. Then, various drug-delivery mechanisms, such as Fickian- and non-Fickian-type diffusion and so on, are shown as fundamental mechanisms (in this paper, this type of diffusion is associated with a cellular or channel-type behavior). In the one-dimensional drug diffusion of a multifractal type, in a controlled-release polymer–drug system in the form of a plane with a certain thickness, a solution is established that allows the model’s validation through the obtained experimental data.

In this paper, new concepts in the treatment of intrauterine adhesions were established, bringing new perspectives for the management of the damaged endometrium. Nevertheless, the mechanism of endometrium regeneration is very complex, and further studies are necessary to investigate the administration technique, the safety, and the short-term efficacy of this drug-delivery system.

## 4. Matherials and Methods

### 4.1. Generalities

The standard models employed for controlled drug release dynamics analyses (such as the zero-order model, Higuchi model, Hixson–Crowell model, etc.) assume different forms of homogeneity (homogenous kinetic space, law of mass, etc.). In this context, differentiable models for describing drug release dynamics become viable. However, such dynamics are usually treated as inherently non-differential (fractal/multifractal). Therefore, the necessity of developing new mathematical models has risen. These models need to be based on the notion of non-differentiability in the description of drug release dynamics, an approach previously not taken into consideration. Thus, non-differentiability allows a “compartmental analysis” in pharmacokinetics, through the modeling of processes such as drug dissolution, absorption, distribution, whole disposition, and the kinetics of bio-molecular reactions [34,35,36]. This type of controlled drug release dynamics can be explicated in the framework of the scale relativity theory (SRT) [37,38,39,40].

In the present paper, by employing the scale relativity theory, some drug-release dynamics are highlighted. 

### 4.2. Towards a Multifractal Pharmacokinetics

Discarding the fractional derivatives and other usual mathematical procedures used in fractal pharmacokinetics [41,42], in the framework of the SRT [38,39,40], the drug-release dynamics can be described through continuous but non-differentiable curves (multifractal curves). In other words, instead of working, for example, with a single variable described by a strict, non-differentiable mathematical function, it is possible to operate only with approximations of this mathematical function. These approximations are obtained by averaging this mathematical function at different scale resolutions, so that any variable proposed for describing drug-release phenomena will still perform as the limit of a family of mathematical functions, this being non-differentiable for null-scale resolution and differentiable otherwise [38,39,40]. Details on the mathematical procedures of the SRT employed in the description of drug-release phenomena are given in Appendix A.

In accordance with Appendix A, one scenario which can be used in the description of drug-release dynamics is the Madelung scenario, that is, the one based on the multifractal hydrodynamics equations system (Equations (A13–A15) from Appendix A):(3a)$$\partial_{t}V_{D}^{i}+V_{D}^{l}\partial_{l}V_{D}^{i}=-\partial^{i}Q$$
(3b)$$\partial_{t}\rho +\partial_{l}\left(\rho V_{D}^{l}\right)=0$$
where
(4a)$$Q=-2\lambda^{2}(dt{)}^{\left[\frac{4}{f\left(\alpha \right)}\right]-2}\frac{\partial^{l}\partial_{l}\sqrt{\rho }}{\sqrt{\rho }}=-V_{F}^{i}V_{F}^{i}-\frac{1}{2}\lambda (dt{)}^{\left[\frac{2}{f\left(\alpha \right)}\right]-1}\partial_{l}V_{F}^{l}$$
(4b)$$V_{D}^{i}=2\lambda (dt{)}^{\left[\frac{2}{f\left(\alpha \right)}\right]-1}\partial^{i}s,\;V_{F}^{i}=i\lambda (dt{)}^{\left[\frac{2}{f\left(\alpha \right)}\right]-1}\partial^{i}\ln\rho$$
(4c)$$\rho =\Psi \overset{-}{\Psi },\;\Psi =\sqrt{\rho }e^{is},\;i=1,2,3$$

The meaning of the above equations and quantities are thoroughly discussed in Appendix A.

Now, the controlled drug-release process implies correlating the release dynamics with the two resolution scales (i.e., the differentiable and the non-differentiable one, respectively) through dynamics synchronization in polymer–drug complex structures. 

Mathematically, this is reducible to the following condition:(5)$$V_{D}^{i}=-V_{F}^{i}$$
in which case the multifractal hydrodynamics equations system proves to be reducible to the multifractal diffusion equation:(6)$$\partial_{t}\rho =\sigma \partial_{l}\partial^{l}\rho$$
with
(7)$$\sigma =\lambda (dt{)}^{\left[\frac{2}{f\left(\alpha \right)}\right]-1}$$
a multifractal diffusion coefficient (dependent on the scale resolution—see Appendix A). The dependence of σ on the singularity spectrum $f\left(\alpha \right)$ of order α, with α dependent on the drug release curves fractal dimension $D_{F}$, $\alpha =\alpha \left(D_{F}\right)$, allows “mimicking” specific behaviours:
(i)Monofractal dynamics, when the drug-release dynamics are characterized by a certain fractal dimension of the drug-release curves. In this way, for drug-release curves with $D_{F}\to 2$ (Peano-type curves [37,38]), Fickian-type drug-release regimes can be “mimed”, while for drug-release curves with $D_{F}\neq 2$, non-Fickian-type drug-release regimes can be “mimed”.(ii)Multifractal dynamics, when the drug-release dynamics are simultaneously characterized by more than one fractal dimension of the drug-release curves. In this way, mixed drug-release regimes (both of a Fickian and non-Fickian type) can be “mimed”. We note that it can be regularly found that $D_{F}\leq 2$ for correlative drug-release modes and $D_{F}>2$ for non-correlative drug-release modes (for details on correlative and non-correlative-type behaviors see [37]).


### 4.3. Drug-Release Regimes through Harmonic Mappings

In accordance with Appendix B, a “hidden” symmetry of a SL(2R)-type [39,40], induced by the multifractal scalar potential
(8)$$\chi =2i\lambda {\left(dt\right)}^{\left[\frac{2}{f\left(\alpha \right)}\right]-1}\ln\Psi$$
of the multifractal velocity field
(9)$${\hat{V}}^{i}=-\partial^{i}\chi$$
allows the “mimicking” of drug-release regimes by correlating the level of release with the scale resolution. The specification of these drug-release regimes implies the following operational procedures:(i)At any resolution scale, a metric of the Lobachevsky plane in the Poincare’s representation [39,40]
(10a)$$\frac{ds^{2}}{k^{2}}=4\frac{dhd\overset{-}{h}}{{\left(h-\overset{-}{h}\right)}^{2}}=\frac{du^{2}+dv^{2}}{v^{2}},$$
(10b)$$h=u+iv,\;\overset{-}{h}=u-iv$$
obtained as a Caylean metric of an Euclidean plane, for which the absoluteness is a circle with unit radius. In this way, the Lobachevsky plane can be put into biunivoc correspondence with the interior side of this circle.

(ii)At any resolution scale, harmonic mappings from the usual space to the hyperbolic one by means of the stationary values of a Lagrangian connected to the metric Equation (10a).(iii)At any resolution scale, field equations (of a Euler–Lagrange-type) connected to the metric Equation (10a)

(11)$$\left(h-\overset{-}{h}\right)\nabla^{2}h=2{\left(\nabla h\right)}^{2}\;\left(h-\overset{-}{h}\right)\nabla^{2}\overset{-}{h}=2{\left(\nabla \overset{-}{h}\right)}^{2}$$ admitting the solution
(12)h=cosh⁡Φ2−sinh⁡Φ2e−iαcosh⁡Φ2+sinh⁡Φ2e−iα, α∈R
with α real and arbitrary and
(13)∇2Φ2=0

(i)At any resolution scale, a transition from stationary to non-stationary states in the polymer–drug dynamics by choosing α=Ωt, in which case Equation (12) takes the form:
(14)$$h=\frac{i\left[e^{2\Phi }\sin\left(2\Omega t\right)-\sin\left(2\Omega t\right)-2ie^{\Phi }\right]}{e^{2\Phi }\left[\cos\left(2\Omega t\right)+1\right]-\cos\left(2\Omega t\right)+1}$$(ii)At any resolution scale, the SL(2R) group, as an invariance group with respect to the metric Equation (10a), operates also as a synchronization group in drug-release dynamics.

Now, let us specify the drug-release regimes, using, for this purpose, the results from Equation (14).

According to observations from Appendix A, the drug-release processes are controlled, for the most part, through correlative release modes, while also being controlled, to a minor extent, through non-correlative release modes (see Figure 7).

At any resolution scale, the correlative release modes are mainly induced by the self-organization processes of the polymer-drug dynamics. 

The selection of scale resolution imposes, however, the drug release regime:(a)through cellular-type structures (see Figure 8a)(b)through channel-type structures (see Figure 8b)(c)through cell-channel-type mixed structures (see Figure 8c). In our opinion, no matter the release regime, these regimes must be matched with polymer–drug network phonon field dynamics.

Moreover, by plotting h in dimensionless coordinates, specific temporal self-similar properties of the polymer–drug release dynamics can be seen in Figure 1, Figure 2, Figure 3 and Figure 4 (see Appendix C). Since the structure’s correlation channel has an exponential decrease in the (Ω;*t*) plane, dissipation processes [39,40,41,42] occurring during drug release are highlighted. This model allows the expression of the drug dissipation through the reduction of the channel amplitude on the Ω axis as the time variable is increased.

From these figures, one can also observe channel-type patterns through the self-structuring of the polymer–drug system entities.

Let us once again note that, in the present context, the scale resolution must be correlated with the rate of drug delivery. The differentiable scale resolution implies drug delivery dynamics at the macroscopic scale, while the non-differentiable scale resolution implies drug delivery dynamics at the microscopic scale, so that the previous figures refer to specific sequences of drug release dynamics.

### 4.4. Materials

Low-molecular-weight chitosan (193 kDa, DD = 82%), 2-hydroxybenzaldehyde (98%), ethanol, acetic acid, phosphate buffer (PBS) (pH = 7.4), and diclofenac sodium salt (DCF) were procured from Sigma-Aldrich (St. Louis, MO, USA) and used as received.

### 4.5. Preparation of the Formulations

Two formulations were prepared by chitosan hydrogelation with 2-hydroxybenzaldehyde [20]. Briefly, a solution of a mixture of 2-hydroxybenzaldehyde and diclofenac sodium salt in ethanol (1%, *w*/*v*) was slowly dropped into a 2% chitosan solution in 0.7% acetic acid, under magnetic stirring at 55 ^°^C. The viscous reaction mixture instantaneously transformed into a yellowish, transparent hydrogel. By varying the ratio between the glucosamine units of chitosan and 2-hydroxybenzaldehyde (1.5/1 and 2/1) while maintaining a constant DCF amount (2.4% from the dry mass of final sample), two formulations were prepared. To provide an insight into the formulations’ compositions, a schematic representation is provided in Figure 9.

### 4.6. Characterization of the Hydrogel Formulations 

The as-obtained hydrogel formulations were lyophilized with a Labconco Free Zone Freeze Dry System instrument (Labconco Corporation, Kansas City, MO, USA) for 24 h at −54 °C and 1.512 mbar to produce the corresponding xerogels, which were subjected to further analysis.

FTIR spectra on the xerogel samples and DCF powder were recorded on a Bruker Vertex 70 Spectrophotometer (Billerica, MA, USA) equipped with a ZnSe single reflection ATR accessory. The spectra were registered in the 600–4000 cm^−1^ spectral range, with 32 scans at 4 cm^−1^ resolution, and processed with OriginPro version 8.0 software.

The hydrogel state has been confirmed by inverted tube test and by rheological investigations on a MCR 302 Anton-Paar rheometer equipped (Anton Paar GmbH, Graz, Austria) with a Peltier device, using a plane-plane geometry (diameter of 50 mm) and an anti-evaporation device, which limits the water evaporation. The loss modulus (G”) and the storage modulus (G’) were determined by the oscillatory deformation tests, at 37 °C.

The morphology of the formulations was assessed via scanning electron microscopy (SEM), by acquiring images on a EDAX-Quanta 200 scanning electron microscope (FEI Company, Hillsboro, OR, USA) −20 kV, WD = 10.1, and 10.9 mm, respectively, and via polarized optical microscopy (POM), by acquiring images on a Leica DM 2500 microscope (Leica, Wetzlar, Germany).

The swelling degree was investigated by immersing square pieces of 10 mg xerogel into vials containing water and weighing the swelled hydrogel at every hour until a constant weight was reached. The swelling degree was calculated as the ratio of the mass of absorbed water to the mass of initial xerogel. The measurements were done in triplicate, and the mean values were given. 

The in vitro release of DCF from the formulations was surveyed under conditions mimicking the physiological environment, in PBS of pH 7.4 at 37 °C, for 10 days. Pellets of xerogel formulations weighing 62 mg each were dipped into vials containing 10 mL of PBS and the vials were placed into an Cryste Puricell 80 incubator (Novapro Co., Ltd., Seokcheon-ro, Bucheon-si, Gyeonggi-do, Republic of Korea) under gentle shacking of 25 rot/min. Furthermore, the concentration of DCF in the supernatant was calculated by measuring the supernatant’s absorbance at 275 nm on a PerkinElmer Lambda 10 (PerkinElmer, Waltham, MA, USA) and fitting it to a previously drawn calibration curve [18,19,20]. The cumulative DCF release at each moment has been calculated by applying the following equation: (15)$$\%DCF=\frac{10C_{n}+2\sum C_{n-1}}{m_{0}}\times 100$$
where $C_{n}$ and $C_{n-1}$ represent the concentrations of DCF in the supernatant after *n* and *n* − 1 withdrawing steps, respectively, and $m_{0}=1.5\;mg$, corresponding to the DCF amount in the initial formulation hydrogels. The measurements were done in triplicate, and the mean values were presented. 

## Figures and Tables

**Figure 1 gels-09-00422-f001:**
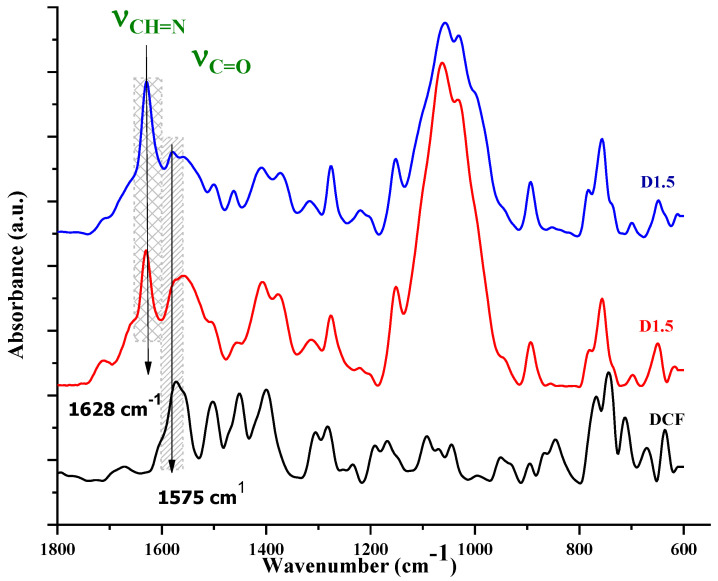
The FTIR spectra of the studied formulations and DCF.

**Figure 2 gels-09-00422-f002:**
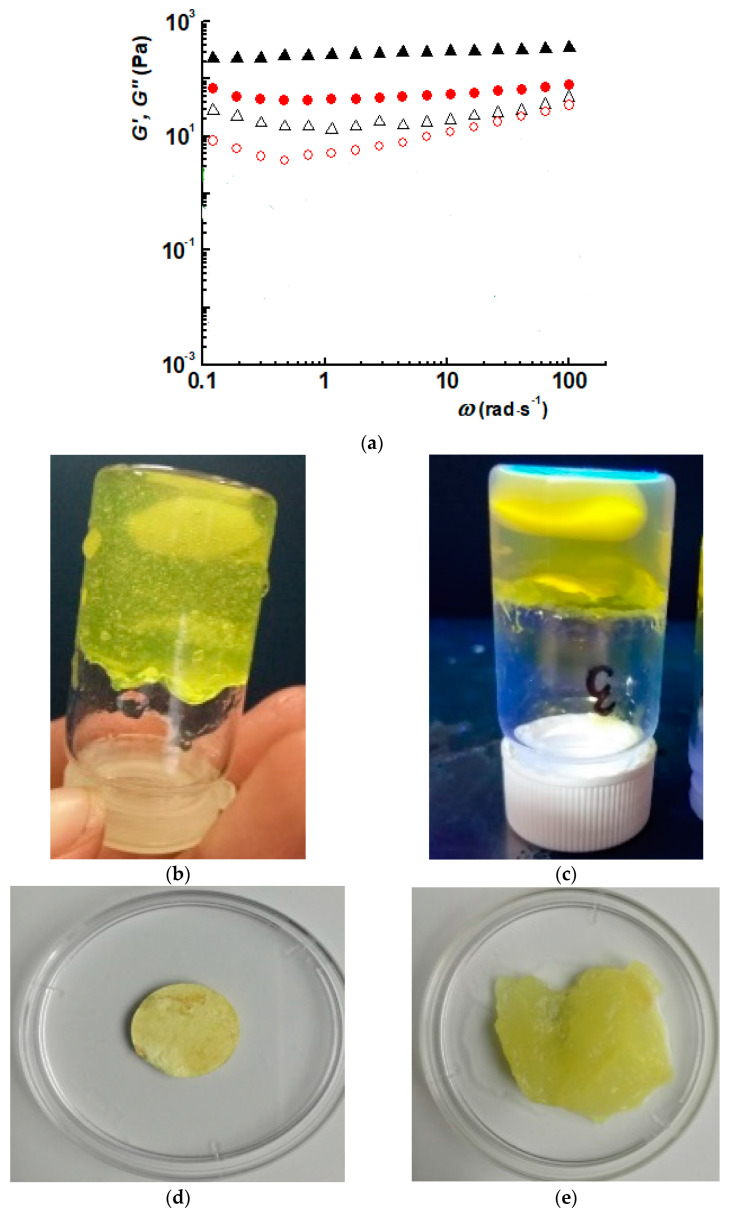
(**a**–**e**) Hydrogel state of the formulations proved by (**a**) the dependence of G’ and G’’ on the oscillatory frequency (black: G’; red: G’’; triangle: D1.5; circle: D2) and (**b**,**c**) inverted tube test; representative image of D1.5 hydrogel, (**d**) before and (**e**) after swelling in water.

**Figure 3 gels-09-00422-f003:**
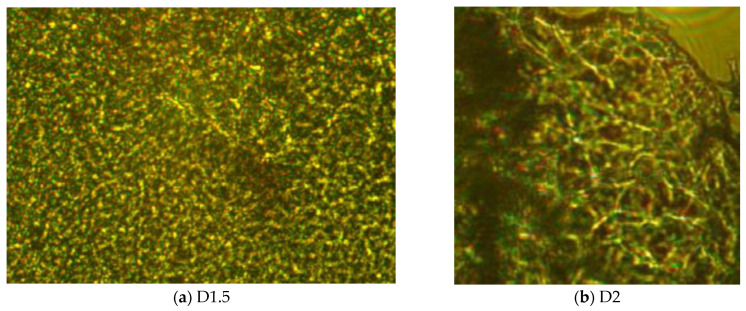
POM microphotographs (crossed polarizers): (**a**) D1.5 and (**b**) D2 formulations (magnification: 400×).

**Figure 4 gels-09-00422-f004:**
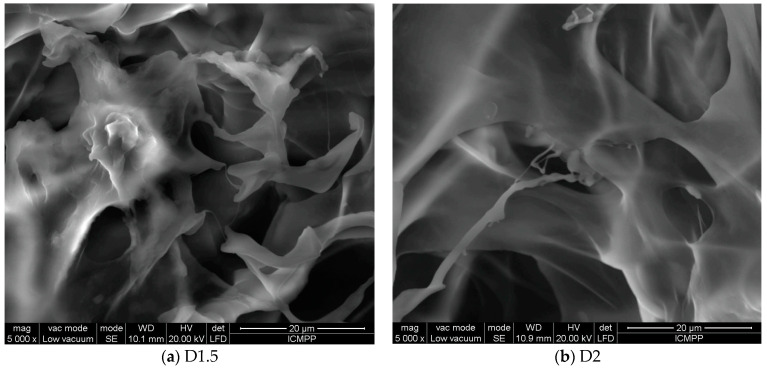
SEM images of the (**a**) D1.5 and (**b**) D2 formulations (magnification: 5000×; scale bar: 20 μm).

**Figure 5 gels-09-00422-f005:**
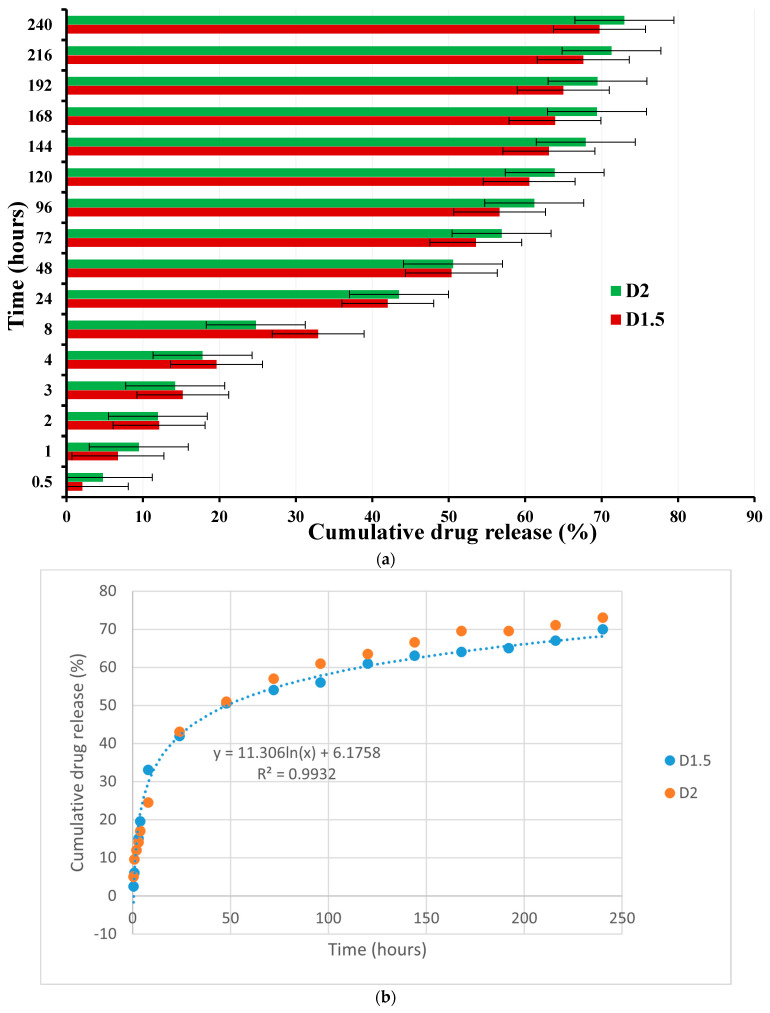
(**a**,**b**) Curves of the in vitro release of DCF from the chitosan-based matrix: (**a**) time (hours)–cumulative drug release (%); (**b**) cumulative drug release (%)–time (hours).

**Figure 6 gels-09-00422-f006:**
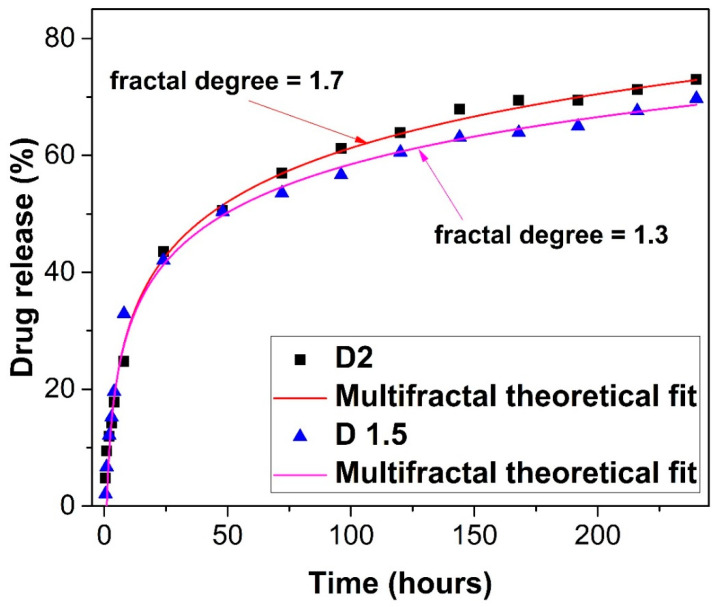
Fractal fit of the in vitro release of diclofenac sodium salt from the chitosan hydrogel matrix. The release curve for the D1.5 formulation corresponds to a fractal degree of 1.3, whereas the release curve for the D2 formulation corresponds to a fractal degree of 1.7, implying the fact that the drug-release processes take place on two resolution scales.

**Figure 7 gels-09-00422-f007:**
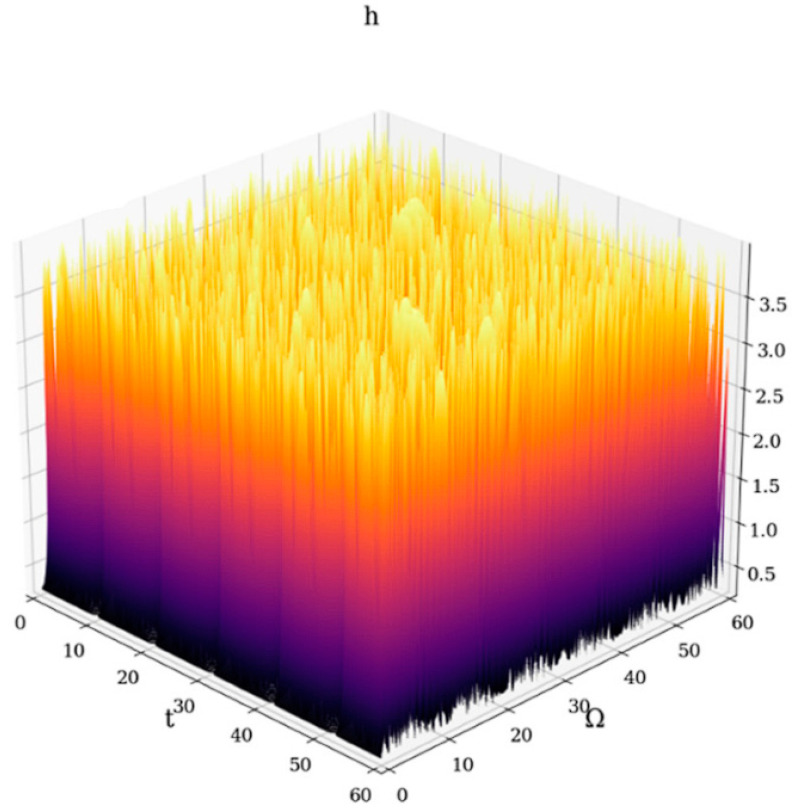
3D representation of drug-release correlative and non-correlative modes at global scale resolution (simultaneously differentiable and non-differentiable scale resolution) plotted in dimensionless coordinates through $\left|h\left(\Omega ,t\right)\right|$ with Φ≡2.35. Such a representation encompasses various types of drug-release processes.

**Figure 8 gels-09-00422-f008:**
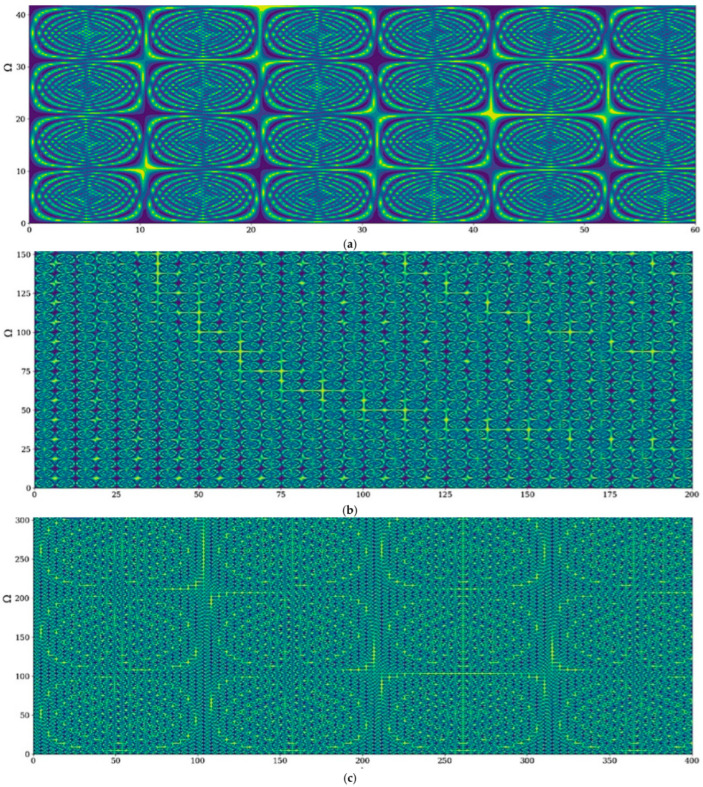
(**a**–**c**) 3D representations of drug-release correlative and non-correlative modes at global scale resolution (simultaneously differentiable and non-differentiable scale resolution) plotted in dimensionless coordinates through $\left|h\left(\Omega ,t\right)\right|$ with Φ≡2.35: (**a**) cellular-type structures (Ω=0−40, t=0−60); (**b**) channel-type structures (Ω=0−150, t=0−200); (**c**) mixed cellular-channel-type structures (Ω=0−300, t=0−400).

**Figure 9 gels-09-00422-f009:**
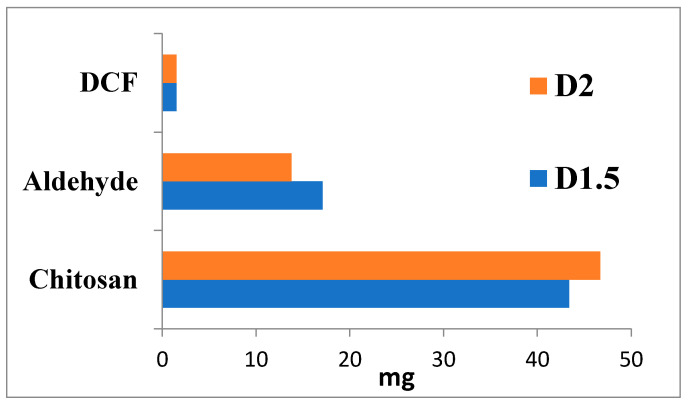
Graphical representation of compositions of the formulations and their codes.

## Data Availability

Not applicable.

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
