# Peer review of "Theoretical and Experimental Aspects of Sodium Diclofenac Salt Release from Chitosan-Based Hydrogels and Possible Applications"

_gels, 2023, doi:10.3390/gels9050422_

Round 1
Reviewer 1 Report
The manuscript “Prevention of Intrauterine Adhesions Using Sodium Diclofenac Salt—Chitosan-Based Hydrogels: Theoretical and Experimental Aspects” is devoted to an actual topic and has a high applied value.
Authors prepared two chitosan-based formulations, containing DCF as a model drug. POM and SEM microscopy indicated that the DCF drug was embedded into the hydrogel pore walls at sub-micrometric level. The in vitro drug release of DCF showed prolonged delivery, reaching 40% in the first 24 h. A new theoretical model to describe drug-delivery processes was proposed.
Minor remarks:
1. I would recommend to the authors to rewrite the abstract. For some reason, the authors singled out subparagraphs 1-3 in it. In addition, the abstract contains many general phrases, authors should also add the results of their own research and conclusions drawn on their basis.
2. Section 3.3 is written very briefly. I recommend expanding the description of all research methods. In addition, it is necessary to add a description of FTIR to this section.
3. Typos:
«In vitro» should be italicized: lines 31, 145, 360, 414, 438.
Line 190: extra bracket.
Figure 9: is missing 9e, apparently, the authors messed something up with Figures 9e and 9f.
Line 427: N in poly(N-vinylcaprolactam) should be italicized
Line 488: in situ should be italicized
Reviewer 2 Report
This Manuscript is adequate for Gel Chemistry and Physics section and also relevant to the field in both theoretical and experimental aspects. Instead, I would like to suggest some recommendations before its publications
- The equation development section is long and could be summarized without prejudice to the reader or replaced in supplementary material;
- Blue frame involving Figures 1-6 could be removed with no loss of quality;
- Specify G´and G´´ in graphic points;
- There is no Equation 41 (see line 442), please check;
- If this (sophisticated) fractal model, when reduced to the real conditions, is identical to the Higuchi model, and, by using Occam's razor, could the authors please clarify the significant advantages of using this model?
Reviewer 3 Report
Dear respected editor
The manuscript entitled “Prevention of Intrauterine Adhesions Using Sodium Diclofenac Salt—Chitosan-Based Hydrogels: Theoretical and Experimental Aspects." reported the preparation and the the drug-release behavior of two chitosan crosslinked hydrogels The drug delivery was monitored in vitro, and a mathematical model was created to understand the release kinetics
The manuscript can be accepted for publications after major corrections
1. First of all the manuscript should be checked by an English native speaker to remove the syntax and typos.
2. The title is not reflecting the work that has been done
3. The introduction is too long, it is preferred to be reduced and to be more concise and to be to the point
4. please adjust the numbering as equation number 2 is missing
5. The authors have written a lot of equation and do not mention the aim of all these equations
6. What is the significance of figures from 1 to 6, is the methodology a suitable place for these figures?
7. The title named 3.3. Equipment and Methods should be changed to characterization of the hydrogel.
8. The figure for calibration curve should be removed from the methodology, it is better to be placed in results and discussion part or to be removed from the whole manuscript and just state the lambda max used for measurement
9. The conditions for the in vitro release should be stated in details
10. The authors should state the rational for using the investigated concentrations in the preparation of the hydrogel; did they depend on preliminary study or previously published work?
11. The SEM figure should include the magnification
12. The manuscript contain too many figures that should be reduced
13. Figure 12 of the in vitro release should be presented in the classic form of time presented on the x axis and the cumulative release presented on the y axis for better understanding of the release pattern
14. Statistical analysis must be performed for the obtained results
15. All figures should present the standard deviation bars
16. No in vitro or in vivo test was conducted to ensure the feasibility of the prepared gel to prevent intrauterine adhesions.
17. Page 22 has a paragraph starting with the “Inflammation plays a key …………………..”
What is the evidence supporting the conclusion in this paragraph. The study only studied the release for the further development of the formulations for clinical applications
Round 2
Reviewer 3 Report
the manuscript can be accepted for publication after adding the missed standard deviation bars to the figures
